# MIXTURE STOCHASTIC BLOCK MODEL FOR MULTI-GROUP COMMUNITY DETECTION IN MULTIPLEX GRAPHS

## ABSTRACT

Multiplex graphs have emerged as a powerful tool for modeling complex data due to their capability to accommodate multi-relation structures. These graphs consist of multiple layers, where each layer represents a specific type of relation. Pillar community detection, a clustering approach that assigns vertices to clusters across all layers, has been employed to identify shared community structures. However, particular layers may possess distinct divisions, deviating from the pillar-based clustering. Consequently, it becomes crucial not to identify individual layer clusters, but a similar cluster for similar layers. In this paper, we propose an approach called the "Mixture Stochastic Block Model," which aims to group similar layers based on shared community structures. A common Stochastic Block Model represents each group's shared community structure. The model is rigorously defined, and an iterative technique is employed for computing the inference. We estimate the layer-to-group assignments using the expectation-maximization technique, while the vertex-to-block assignments within each group are determined using the variational estimation-maximization technique. We assess the identifiability of our proposed model and show the consistency of the maximum likelihood function. The performance of the method is evaluated using both synthetic graphs and real-world datasets, showing its efficacy in identifying consistent community structures across diverse multiplex graphs.

## 1 INTRODUCTION

In recent times, rapid advancements in technology have led to an exponential increase in the accumulation of data. This has ushered in the era of big data, which poses new challenges in terms of exploring and analyzing vast quantities of information Elgendy & Elragal (2014). The data is often presented from multiple perspectives, where various phenomena of interest can be explained through diverse sources with multiple features Devagiri et al. (2021); Niu et al. (2016). To cope with the growing complexity, the concept of multiplex graphs has emerged as a valuable tool in this context Hammoud & Kramer (2020); Zweig (2016).

A multiplex graph comprises a collection of interconnected vertices shared across different layers Han et al. (2023); Magnani et al. (2021). It serves as an effective means of representing multi-relational data, where a distinct set of edges represents each feature. Moreover, the multiplex graph demonstrates considerable advantages when data features may dynamically change over time Xia et al. (2020). This flexibility allows for exploring and analyzing time-varying data with enhanced precision and adaptability.

The application of clustering has proven to be an effective means of understanding and exploring data in various fields Fortunato (2010a); Plantié & Crampes (2013); Bedi & Sharma (2016), by identifying set of individuals who exhibit strong similarities. However, traditional clustering techniques encounter challenges, particularly the curse of dimensionality, when dealing with datasets with many features Sisodia et al. (2012). To address this limitation, the multiplex graph offers a promising solution Wang et al. (2019) such that the communities detection on such graph aims to identify groups with high intra-connectivity and low inter-connectivity Fortunato (2010b). Numerous algorithms have been developed for community detection in multiplex graphs, utilizing various

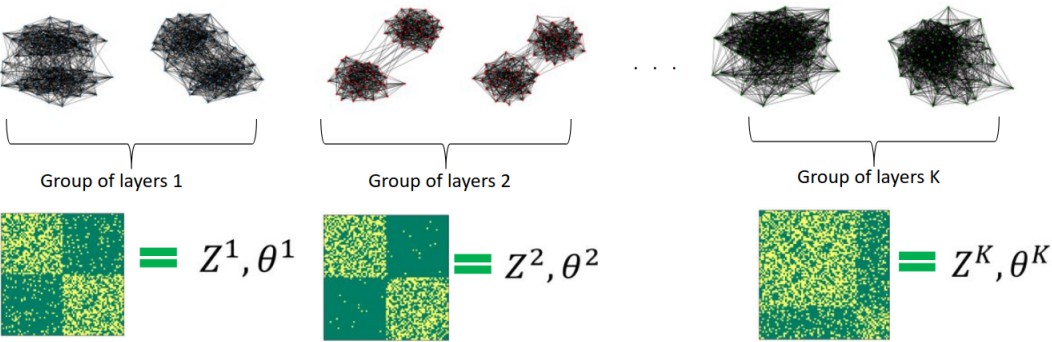

Figure 1: From the multiplex graph, the MSBM model identifies similar layers into groups. For each group, a specific SBM is estimated, such that the edges are divided into blocks.

approaches such as optimization De Meo et al. (2011); Que et al. (2015), spectral computation Li et al. (2018), consensus clustering Mandaglio et al. (2018), and inference Shuo & Chai (2016). However, those methods generally provide a single cluster over all layer, which may not reveal the quality and diversity of alternative clusters of the multiplex graph. Especially, the layers may exhibit a complementary information with different structural clustering, like the communities of a given user in social preferences which may be different between music and movies preferences. Such diversity leads us to explore the identification of similar layer into groups from where a specific division of each group will be proposed.

The Stochastic Block Model (SBM) Lee & Wilkinson (2019) is a generative model that represents a widely adopted technique for community detection. The SBM considers the existence of communities within the graph and characterizes the probability of edges between vertices based solely on the communities to which the connecting vertices belong. It is versatile enough to be applied across various type's graphs, including mono-layer Abbe (2017), multiplex graph Barbillon et al. (2017), and dynamic networks Corneli (2017). For the multiplex graph, the SBM has been used to perform actor-based clustering Barbillon et al. (2017), which has been extensively utilized to group vertices consistently within all layers of the multiplex graph. However, this modeling can not capture the diversity of clusters withing the multiplex graph, that may results in sub-optimal communities.

In this paper, we propose Mixture SBM (MSBM) for multi-group clustering, such that **the layers with similar clustering are joined into groups**, and **the vertices of each group are clustered into blocks**, as shown in the figure 1. To address the double clustering, we assume that the layers of the same group shares identical SBM distribution. Estimation Maximization (EM) algorithm is used to identify the group of layers and the Variational Estimation Maximization (VEM) technique is used to assign vertices to their respective blocks in each group.

The paper is organized by showing a brief related work in Section 2. We set the model with its variables in Section 3. Then, we explain the different steps of optimization and we introduce the model in which it is used for initialization the latent variables in Section 4. We then pass to the experiment part in Section 5, and finish the paper by a conclusion in Section 6.

## 2 RELATED WORKS

The application of community detection for exploring and comprehending complex data through multiplex representation has emerged as a highly compelling and active research domain. Various algorithms have been developed for this purpose and can be broadly categorized into three groups.

The first category employs a flattening technique, where a mono-layer representation of the multiplex graph is computed to summarize its structure. Therefore, community detection is performed using traditional mono-graph algorithms on the flattened layer Berlingerio et al. (2011); Kim et al. (2016). However, this approach may only partially capture the inter-layer affinity, where noisy layers can adversely affect the clustering outcomes.

The second category of methods focuses on computing consensus clustering across all layers of the multiplex graph. This approach entails determining individual groups for each layer using mono-layer algorithms and then optimizing a division that maximizes consensus across all layers Berlingerio et al. (2013); Tagarelli et al. (2017); Tang et al. (2012). However, it is essential to note that while reducing the graph to clusters, some of the rich information in the original graph may need to be recovered.

The third category is the direct methods, where algorithms aim to optimize communities directly from the information provided by the whole graph Papalexakis et al. (2013). Many mono-layer algorithms have been extended to handle multiplex graphs De Domenico et al. (2015); Boutemine & Bouguessa (2017); Afsarmanesh & Magnani (2016); Mucha et al. (2010), and the Multi-Layer Stochastic Block Models (MLSBMs) is one such extension Vallè s-Català et al. (2016); De Bacco et al. (2017); Paul & Chen (2016). MLSBMs assume the existence of community patterns between the layers and seek to capture layer similarity using a SBM model. These models infer the graph's structure using techniques like Variational Estimation Maximization Barbillon et al. (2017); Celisse et al. (2012); Corneli et al. (2016); Paul & Chen (2016); Han et al. (2015). However, such models suffer from an exponential parameter increase and may struggle to identify different clustering patterns between the layers. A related work described in Stanley et al. (2015) proposes a model with multi-division for a multiplex graph. This model infers multiple SBMs for the multiplex graph, where each SBM corresponds to a specific community structure. The similar layers present their groups, with their SBMs generated independently. However, this approach uses the K-means algorithm on the parameters for identifying layers within the same groups, which can be limiting as K-means has serious limitation like its hypothesis to be suited for spherical topology.

In this paper, we present a novel mixture model that jointly represents the affiliation of each layer to a group (set of layers), and vertices within each group are assigned to a single block (set of vertices). We employ the EM-VEM technique to infer the model's parameters and estimate the assignment variables effectively, offering an improved approach for community detection in multiplex graphs.

## 3 MIXTURE STOCHASTIC BLOCK MODEL

This work aims to establish a joint clustering of similar layers into groups, and for each group, a single clustering of vertices into blocks. Therefore, layer-to-groups variables are estimated to identify the group of each layer. The edges within each group are assumed to follow an unique SBM distribution, such that the layer of the same group are considered as samples from the same distribution. Additionally, for each group, the vertex-to-block variables are to identify the block of each vertex within the same group.

### 3.1 MODEL DEFINITION

Consider a multiplex graph denoted as $\mathcal{G} = \{G^1, ...G^L\}$ comprising $L$ layer, with $G^l = \{V, E^l\}$ represents a single layer, where $l$, s.t $l \in [1, L]$ indicates the layer index, $V$ indicates the set of vertices with $|V| = N$, $E^l$ the set of edges within layer $l$. Let $\mathcal{A} = \{A^1, ..., A^L\}$ be the corresponding adjacency matrix of multiplex graph $\mathcal{G}$, where $A^l$ stands for the adjacency matrix of the graph $G^l$. The underlying graph model in this study is an unweighted and undirected multiplex graph, where the edge distributions follow a Bernoulli distribution. The generalization of this model to a directed graph is straightforward. Finally, an edge $A^l_{ij}$ is defined by a dyad representing its extremity $i, j$.

Let's consider a partition of multiplex's graph layers into $K$ groups and assume that the vertices of group $k$ are divided into $Q^k$ clusters that we names blocks, where $k \in [1, K]$. The probability of having an edge $A^l_{i,j}$ in layer $l$, within group $k$, giving the block assigned to each vertex, is expressed as follows:

$$P(A^l_{i,j}|\mathbf{Z}^k, \mathbf{\Pi}^k) = \pi^k_{Z^k_i, Z^k_j} \tag{1}$$

where $\mathbf{Z}^k = \{Z^k_1, ...Z^k_N\}$ is the set of vertex-to-block assignments in the group $k$ and $Z^k_i \in \{1, ..., Q^k\}$. The matrix $\mathbf{\Pi}^k$ has $Q^k \times Q^k$ elements $\pi^k_{q,w}, \forall q, w \in \{1, ..., Q^k\}^2$. Each element represents the probability of an edge existing, depending on the block of its dyad.

Let consider a distinct layer $G^l$, with the vertex-to-block variables assignment for each group, where $\mathbf{Z} = \{\mathbf{Z}^1, ..., \mathbf{Z}^K\}$, the probability of existing edge $A_{ij}^l$ between dyad $(i, j)$ from the MSBM model, conditioned on $\mathbf{Z}$, can be described as a mixture distribution of $K$ independent SBMs, expressed as:

$$P(A_{ij}^l = 1|\mathbf{Z}; \boldsymbol{\beta}, \mathbf{\Pi}) = \sum_{k=1}^{K} \beta^k \pi_{Z_i^k, Z_j^k}^k$$

$$s.t \sum_k \beta^k = 1 \tag{2}$$

where $\boldsymbol{\beta} = \{\beta^1, ..., \beta^K\}$ is the set of probability of layer $l$ to be generated from the group $k$, which represents the mixture weights for MSBM, and $\mathbf{\Pi} = \{\mathbf{\Pi}^1, \mathbf{\Pi}^2, ..., \mathbf{\Pi}^K\}$ is the set of SBM parameters of each group. This model accounts for the incorporation of $K$ distributions from which layers can be generated. To address the challenge of maximizing the log-likelihood function due to the sum in the mixture model, we introduce a set of latent variables $\mathbf{Y}$ that represents the layer-to-group assignment. Specifically, $y_{lk}$, s.t $l \in [1, L]$ and $k \in [1, K]$, takes the value of one when layer $l$ is generated from group $k$ and zero otherwise. The updated formulation for the probability of an existing edge is as follows:

$$P(A_{ij}^l = 1|\mathbf{Y}, \mathbf{Z}; \mathbf{\Pi}) = \prod_{k=1}^{K} (\pi_{Z_i^k, Z_j^k}^k)^{y_{lk}}$$

$$P(y_{lk} = 1; \beta) = \prod_{k=1}^{K} (\beta^k)^{y_{lk}} \tag{3}$$

Additionally, for any group $k$, we identify the probability of a vertex $i$ to be assigned to block $q$ as follows:

$$P(Z_i^k = q; \boldsymbol{\alpha}^k) = \alpha_q^k$$

$$s.t \sum_{q=1}^{Q^k} \alpha_q^k = 1 \tag{4}$$

such that $\boldsymbol{\alpha} = \{\boldsymbol{\alpha}^1, \boldsymbol{\alpha}^2, ..., \boldsymbol{\alpha}^K\}$ and $\boldsymbol{\alpha}^k = \{\alpha_1^k, \alpha_2^k, ..., \alpha_{Q_k}^k\}$.

Let us consider $\boldsymbol{\theta} = \{\boldsymbol{\theta}^1, \boldsymbol{\theta}^2, ..., \boldsymbol{\theta}^K\}$, such that $\boldsymbol{\theta}^k = \{\mathbf{\Pi}^k, \boldsymbol{\alpha}^k\}$, the log-likelihood of the proposed model is written as follows:

$$\mathcal{L}(\mathcal{A}, \mathbf{Y}, \mathbf{Z}; \boldsymbol{\beta}, \boldsymbol{\theta}) = \sum_{l=1}^{L} \sum_{k=1}^{K} y_{lk} \left[ ln\beta^k + \mathcal{L}(A^l, \mathbf{Z}^k; \boldsymbol{\theta}^k) \right] \tag{5}$$

where $\mathcal{L}(A^l, \mathbf{Z}^k; \boldsymbol{\theta}^k)$ is the complete log-likelihood of layer $l$ in group $k$ with parameters $\boldsymbol{\theta}^k$, formulated as follow:

$$\mathcal{L}(A^l, \mathbf{Z}^k; \boldsymbol{\theta}^k) = ln(P(A^l|\mathbf{Z}^k; \mathbf{\Pi}^k)) + ln(P(\mathbf{Z}^k; \boldsymbol{\alpha}^k))$$

$$= \sum_{i,j,i \neq j} A_{ij}^l ln(\pi_{Z_i Z_j}^k) + (1 - A_{ij}^l) ln(1 - \pi_{Z_i Z_j}^k) + \sum_{i=1} ln(\alpha_{Z_i}^k) \tag{6}$$

The verification of the parameter identifiability and the assessment of the maximum likelihood consistency have been performed in supplementary materials.

In the context of inferring information from a given multiplex graph, the primary objectives involve assigning each layer to a specific group $k$ using variable $y_{lk}$, then assigning each vertex $i$ within group $k$ to a particular block $q$ using variable $Z_i^k$, and optimizing the parameters $\boldsymbol{\beta}$ and $\boldsymbol{\theta}$.

# 4   OPTIMIZATION OF LOG LIKELIHOOD FUNCTION

As explained previously, the MSBM depends on layer-to-group and vertex-to-block assignment variables. We set an iterative approach to address this jointly assignment clustering challenges. To elaborate, we start with the estimation of layer-to-group assignment variables by utilizing the Estimation Maximization (EM) algorithm. Then, to infer the SBM representation within each group, we use the Variational EM (VEM) technique. This technique proves its performance in maximizing SBM distribution parameters while estimating the latent vertex-to-block variables.

## 4.1   ESTIMATION OF LAYER-TO-GROUP VARIABLES

The computation of layer-to-group latent variables estimation is derived from equation 5, which defines the complete log-likelihood. The estimation process involves calculating the expectation of the log-likelihood based on the posterior distribution of layer-to-group latent variables, and it can be expressed as follows:

$$E_{\mathbf{Y}}[\mathcal{L}(\mathcal{A}, \mathbf{Y}, \mathbf{Z}; \boldsymbol{\beta}, \boldsymbol{\theta})] = \sum_{l=1}^{L} \sum_{k=1}^{K} E(y_{lk}) \Big[ ln\beta^k + \mathcal{L}(A^l, \mathbf{Z^k}; \boldsymbol{\theta}^k) \Big] \tag{7}$$

where $E(y_{lk})$ is the posterior expectation probability of layer $l$ to be generated from group $k$, defined as $p(y_{lk}|A^l, \mathbf{Z}^k)$. Using Bayes theorem, the estimation of layer-to-group is computed as follows:

$$E(y_{lk}) = \frac{\beta^k P(A^l, \mathbf{Z}^k|\boldsymbol{\theta}^k)}{\sum_j \beta^j P(A^l, \mathbf{Z}^j|\boldsymbol{\theta}^j)} \tag{8}$$

where $P(A^l, \mathbf{Z}^k; \boldsymbol{\theta}^k)$ is written as follows:

$$P(A^l, \mathbf{Z}^k; \boldsymbol{\theta}^k) = P(A^l|\mathbf{Z}^k; \boldsymbol{\Pi}^k) P(\mathbf{Z^k}; \boldsymbol{\alpha^k})$$

$$P(A^l, \mathbf{Z}^k; \boldsymbol{\theta}^k) = \prod_{i,j,i \neq j} (\pi_{Z_i Z_j}^k)^{A_{ij}^l} (1 - \pi_{Z_i Z_j}^k)^{(1-A_{ij}^l)} \prod_{i=1}^{N} \alpha_i^k \tag{9}$$

The estimation of layer-to-group prioritizes the layer that maximizes the likelihood distribution for a specific group. In order to assign each layer to a single group, the selection of the group is based on the following:

$$y_{lk} = \underset{j}{\text{argmax}}\, y_{lj} \tag{10}$$

## 4.2   MAXIMIZATION OF LIKELIHOOD PARAMETERS AND VERTEX-TO-BLOCK VARIABLES

Once the layer-to-group assignment is identified, the MSBM parameters can be maximized and vertex-to-group variable can be estimated too.

### 4.2.1   MAXIMIZATION OF $\beta$

Considering equation 5, the optimization of $\boldsymbol{\beta}$ involves expressing the complete log-likelihood as follows:

$$\mathcal{L}(\mathcal{A}, \mathbf{Y}; \boldsymbol{\beta}, \boldsymbol{\theta}) = \sum_{l=1}^{L^k} ln\beta^k + C(\beta^k)$$

$$s.t \sum_{k=1}^{K} \beta^k = 1 \tag{11}$$

where $L^k = \{l \in [1, L], s.t\ \ y_{lk} = 1\}$, set of layer of group $k$, and $C(\beta^k)$ defined as a constant regarding on $\beta^k$. By employing the Lagrange multiplier approach, the solution that satisfies the Karush-Kuhn-Tucker (KKT) conditions can be expressed as follows:

$$\beta^k = \frac{N^k}{N} \tag{12}$$

where $N_k = |L^k|$ is the number of layers in the groups $k$.

### 4.2.2 ESTIMATION OF VERTEX-TO-BLOCK AND MAXIMIZATION OF PARAMETER $\theta_k$

Our mathematical models assume that the layers within the same group are generated independently from SBM distribution specific to that group. Let $\mathcal{A}^k$ the multiplex graph of group $k$, based on equation 6, the log-likelihood of group $k$ is expressed as follows:

$$L(\mathcal{A}^k, \mathbf{Z}^k; \boldsymbol{\theta}^k) = \sum_{l \in L^k} \sum_{i,j,i \neq j} A_{ij}^l ln(\pi_{Z_i Z_j}^k) + (1 - A_{ij}^l)ln(1 - \pi_{Z_i Z_j}^k) + \sum_{i=1} ln(\alpha_{Z_i}^k)$$

$$s.t \sum_{q}^{Q^k} \alpha_q^k = 1 \tag{13}$$

In order to optimize the parameters that maximize the previous equation, it is essential to first estimate the latent assignment variables. This task is addressed using the Estimation Maximization (EM) algorithm, which requires computing the posterior probability of the latent variable $\mathbf{Z}^k$ with respect to the observed layers, denoted as $P(\mathbf{Z}^k|\mathcal{A}^k)$. However, for single-layer graphs, it has been demonstrated that calculating this conditional probability is computationally intractable Celisse et al. (2012). Various approaches have been proposed in the literature to tackle this challenge Li et al. (2015); Lee & Wilkinson (2019), but they tend to suffer from the curse of dimensionality, mainly when dealing with large-scale datasets.

The Variational EM technique has been adopted to address this issue as an alternative technique for handling SBM estimation challenges. Previous studies have established the VEM technique's convergence for single-layer SBM graphs and multiplex SBM graphs Celisse et al. (2012); Barbillon et al. (2017). The VEM approach involves approximating the posterior distribution $P(\mathbf{Z}^k|\mathcal{A}^k)$, by another distribution $R_{\mathcal{A}^k}$ over $Z^k$. By leveraging this approximation, the marginal log-likelihood over $\mathbf{Z}^k$ can be expressed as follows:

$$\mathcal{L}(\mathcal{A}^k; \boldsymbol{\theta}^k) = \sum_{\mathbf{Z}^k} R_{\mathcal{A}^k}(\mathbf{Z^k})\mathcal{L}(\mathcal{A}^k, \mathbf{Z}^k; \boldsymbol{\theta}^k) - \sum_{\mathbf{Z}^k} R_{\mathcal{A}^k}(\mathbf{Z^k})ln\Big(R_{\mathcal{A}^k}(\mathbf{Z^k})\Big) + \mathbf{KL}\big[R_{\mathcal{A}^k}(\mathbf{Z^k}), P(\mathbf{Z}^k|\mathcal{A}^k; \boldsymbol{\theta}^k)\big] \tag{14}$$

where $\mathbf{KL}$ is the Kullback-Leibler divergence. Therefore, instead of maximizing $\mathcal{L}(\mathcal{A}^k; \theta^k)$ for the observed data, the VEM technique optimizes a lower bound of $\mathcal{L}(\mathcal{A}^k; \theta^k)$, denoted as $\mathcal{I}_\theta(R_{\mathcal{A}^k})$. This lower bound is known as the evidence lower bound, and it can be defined as follows:

$$\mathcal{I}_\theta(R_{\mathcal{A}^k}) = \mathcal{L}(\mathcal{A}^k; \boldsymbol{\theta}^k) - \mathbf{KL}\big[R_{\mathcal{A}^k}(\mathbf{Z^k}), P(\mathbf{Z}^k|\mathcal{A}^k; \boldsymbol{\theta}^k)\big]$$

$$= \sum_{\mathbf{Z}^k} R_{\mathcal{A}^k}(\mathbf{Z^k})\mathcal{L}(\mathcal{A}^k, \mathbf{Z}^k; \boldsymbol{\theta}^k) - \sum_{\mathbf{Z}^k} R_{\mathcal{A}^k}(\mathbf{Z^k})logR_{\mathcal{A}^k}(\mathbf{Z^k}) \tag{15}$$

$$\leq \mathcal{L}(\mathcal{A}^k, \mathbf{Z}^k; \boldsymbol{\theta}^k)$$

The equality between the evidence lower bound and the log-likelihood holds when $R_{\mathcal{A}^k}(\mathbf{Z^k})$ is equal to the true posterior distribution $P(\mathbf{Z}^k|\mathcal{A}^k; \boldsymbol{\theta}^k)$. Maximizing the lower bound $\mathcal{I}_{\boldsymbol{\theta}}(R_{\mathcal{A}^k})$ is equivalent to minimizing the Kullback-Leibler divergence $\mathbf{KL}\big[R_{\mathcal{A}^k}(\mathbf{Z^k}), P(\mathbf{Z}^k|\mathcal{A}^k; \boldsymbol{\theta}^k)\big]$. Regarding to integer nature of vertex-to-block variables, to approximate the posterior distribution, we select $R_{\mathcal{A}^k}(\mathbf{Z^k})$ as follows:

$$R_{\mathcal{A}^k}(\mathbf{Z^k}) = \prod_{i=1}^{N} h(\mathbf{Z}_i^k; \tau_i^k) \tag{16}$$

where $h(::; \tau_i^k)$ is a multinomial distribution with parameters $\boldsymbol{\tau} = \{\tau_1^k, ...\tau_{Q^k}^k\}$. The entity $\tau_{iq}^k$ approximates the probability that vertex $i$ belongs to the community $q$ in group $k$. The $\mathcal{I}_\theta(R_{\mathcal{G}^k})$ can be writhed as follows:

---

**Algorithm 1** Inference of Likelihood of Multi-Group Stochastic Block Model

---

**Input**: $\mathcal{G}, K, \mathbf{Q} = [Q^1, ..., Q^K]$
**Output**: $\mathbf{Y}, \mathbf{Z}, \mathbf{\Pi}, \boldsymbol{\beta}, \boldsymbol{\alpha}$
  Initialize $\mathbf{Y}, \mathbf{Z}$ with 2
  **while** Iteration $<$ Iteration max $\wedge$ Not Converge **do**
    Estimate $y_{lk}$ with 8
    Compute $y_{lk}$ with 10
    Compute $\alpha_q^k$ with 18
    Compute $\pi_{qw}^k$ with 19
    Compute $\tau_{qw}^k$ with 20
  **end while**

---

$$\mathcal{I}_\theta(R_{\mathcal{A}^k}) = \sum_{l \in L^k} \sum_{i \neq j} \sum_{qw} \tau_{iq}^k \tau_{jw}^k \left[ A_{ij}^l ln(\pi_{qw}^k) + (1 - A_{ij}^l) ln(1 - \pi_{qw}^k) \right]$$
$$- \sum_i \sum_q \tau_{iq}^k ln(\tau_{iq}^k) + \sum_i \sum_q \tau_{iq}^k ln(\alpha_q^k) \tag{17}$$

The parameters that maximize $\mathcal{I}_\theta(R_{\mathcal{A}^k})$ are derived directly from the previously presented formula. To ensure that the vector $\boldsymbol{\alpha}^k$ and matrix $\mathbf{\Pi}^k$ satisfy the constraints $\sum_q \alpha_q^k = 1$ and $0 \leq \pi_{qw} \leq 1, \forall q, w \in \{1, ..., Q^k\}^2$, Lagrange multipliers are employed. The optimal parameters are computed as follows:

$$\hat{\alpha}_q^k = \sum_i \frac{\tau_{iq}^k}{N} \tag{18}$$

$$\hat{\pi}_{qw}^k = \frac{\sum_{l \in L^k} \sum_{i \neq j} \tau_{iq}^k \tau_{jw}^k A_{ij}^l}{\sum_{l \in L^k} \sum_{i \neq j} \tau_{iq}^k \tau_{jw}^k} \tag{19}$$

$$\hat{\tau}_{iq}^k \propto \hat{\alpha}_q^k \prod_{l \in L^k} \prod_{i \neq j} \prod_w \left[ \hat{\pi}_{qw}^{k\ A_{ij}^l} + (1 - \hat{\pi}_{qw}^k)^{(1 - A_{ij}^l)} \right]^{\hat{\tau}_{jw}^k} \tag{20}$$

where $\hat{\boldsymbol{\alpha}}^k, \hat{\mathbf{\Pi}}^k, \hat{\boldsymbol{\tau}}^k$ are the best current parameters. Due to the interdependence between $\hat{\Pi}^k$ and $\hat{\tau}^k$, an effective way to determine the best estimation is to alternate between updating $\hat{\Pi}^k$ and $\hat{\tau}^k$ iteratively until convergence. The optimized parameters define the distribution of SBM and vertex-to-block assignments for a group $k$. The same computation is executed for each group independently. The overall method is summarized in the algorithm 1.

## 4.3 MODEL INITIALIZATION

The initialization process of MSBM involves setting up the layer-to-group variables $\mathbf{Y}$ and the vertex-to-block variables $\mathbf{Z}$. Effective initialization of these assignment variables contributes to faster convergence and a higher chance of recovering accurate ground truth values. In the context of mixture models, the K-means algorithm is commonly employed for initializing assignment variables due to its simplicity and quick response. In this paper, we introduce a novel spectral technique that computes both layer-to-group and vertex-to-block variables, such that the results are used as an initialization for inferring the MSBM model.

Consider $\mathbf{U} = \{\mathcal{U}^1, \mathcal{U}^2, ..., \mathcal{U}^K\}$ set of centroid graphs, with each graph $\mathcal{U}^k$ being the centroid that represents the group $k$. We aim to find layer-to-group variables by optimizing centroids that best represent each group. Then, each centroid $\mathcal{U}^k$ gets clustered into $Q_k$ community for the vertex-to-block assignment variable. One way to find the communities of centroid $k$ is to ensure that it is composed of $Q^k$ disconnected components, where each component corresponds to a community in the graph. In network theory, a graph with a $Q^k$ component exhibits a multiplicity of $Q^k$ null

eigenvalues in its corresponding Laplacian matrix. These null eigenvalues are the smallest eigenvalues of the Laplacian matrix. Thus, minimizing the $Q^k$ smallest eigenvalue of the Laplacian matrix associated with $\mathcal{U}^k$ facilitates the formation of $Q^k$ disconnected components within the centroid. Therefore, the model aiming to optimize these representations can be formulated as follows:

$$
\min_{\mathcal{U}^1, \mathcal{U}^2, \dots \mathcal{U}^K, \mathbf{F}, \mathbf{Y}} \sum_{l=1}^{L} \sum_{k=1}^{K} y_{lk} ||\mathcal{U}^k - \mathcal{A}^l||_F^2 + 2\lambda \sum_{k=1}^{K} Tr(\mathbf{F^k}^T \mathbf{L}_{U^k} \mathbf{F^k})
$$

$$
s.t \ \ \forall i, u_{ij}^k \geq 0, \mathbf{1}^T \mathbf{u}_i^k = 1, \forall k, (\mathbf{F^k})^{\mathbf{T}} \mathbf{F^k} = \mathbf{I}, y_{lk} \in \{0,1\}, \sum_{k=1}^{K} y_{lk} = 1
$$

$$(21)$$

where $||.||_F^2$ denote the Frobenius norm, and $u_{ij}^k$ represent element of the centroid $\mathcal{U}^k$, where $\forall i, j \in V$. The Laplacian representation of centroid $\mathcal{U}^k$ is denoted by $L_{\mathcal{U}^k}$, and $\mathbf{F}^k$ represents an embedding vector.

The Laplacian matrix is computed in its unnormalized version as follows:

$$
L_{\mathcal{U}^k} = D_{\mathcal{U}^k} - \mathcal{U}^k \tag{22}
$$

where $D_{U^k}$ denotes the degree matrix, which is a diagonal matrix. $\mathbf{F}^k \in \mathbf{R}^{N \times Q^k}$ helps to get the number of connected components in the graph. The embedding vector is included in the cost function to soft the constraint of constructing a centroid with $Q^k$ components that effectively represent the communities, leveraging the number of null eigenvalues corresponding to the number of components in the graph. The optimization process for this model are described in the supplementary materials. Experimentally, this initialization helps the MSBM to converge faster than random initialization, up to more than 10 times faster, which depends generally on data structure.

## 4.4 Model Selection

To determine the optimal number of groups $K$ and the number of blocks $Q^k$ in each group $k$, we propose using the Bayesian Information Criterion (BIC). The BIC is formulated as follows:

$$
BIC = 2 \sum_{k=1}^{K} (Q^k)^2 \log(|L^k|N(N-1)) + (Q^k - 1) \log(|L^k|N) + 2(K-1) \log(L) - \mathcal{L}(\mathcal{G}, \mathbf{Y}, \mathbf{Z}; \boldsymbol{\beta}, \boldsymbol{\theta})
$$

$$(23)$$

In this expression, $|L^k|$ represents the number of layers associated with group $k$. Although the theoretical justification for the BIC criterion still needs to be developed Bhat & Kumar (2010), empirical evidence suggests that it yields satisfactory results in practice.

## 5 Experiments

To assess the properties of the MSBM, we compare its performance with various algorithms on four synthetic datasets. Due to space limits, only one experiment is presented in this section, the others are shown in supplementary materials. The results of MSBM on a real datasets are provided in supplementary materials. Specifically, as MSBM performs joint clustering of layer and vertices, we perform separate comparisons between MSBM and other algorithms regarding graph and vertex clustering. For the vertex clustering task, the SBM is the most natural algorithm to compare with. In this context, we employ the SBM across all layers, such that one clustering is returned for all layers. Additionally, We compare MSBM with Generalized Louvain Mucha et al. (2010) and Graph Fusion Spectral Clustering (GFSC) Kang et al. (2019). Those algorithms does not consider the multigroup clusters. Graph clustering is a more complex task. In order to highlight its difficulty, we carry out a comparison with the K-means algorithm, whose hypothesis framework does not necessarily include the complexity of this task. The Normalized Mutual Information (NMI) and the Adjusted Mutual Information (AMI) are used to evaluate the performance of jointly clustering tasks between the algorithms.

## 5.1 SYNTHETIC DATA: VARIABILITY IN BLOCK SIZE

In this experiment, the dataset is composed of 3 groups. Each group of layers consists of 10 graphs, each with 100 vertices organized into four blocks. Vertices inside a block are randomly linked to each other with a probability $\pi_{\text{intra}} = 0.5$, and vertices from different blocks are randomly connected with a probability $\pi_{\text{inter}} = 0.3$. What distinguishes the groups of layers is the number of vertex in each block, $G^1 = \{25, 25, 25, 25\}$, $G^2 = \{20, 25, 25, 30\}$ and $G^3 = \{30, 30, 20, 20\}$, where $G^i$ is the $i^{th}$ group, and for each group, the first number indicates the number of vertex of the first block, the second number the one of the second block, and so forth, as shown in figure **??**.

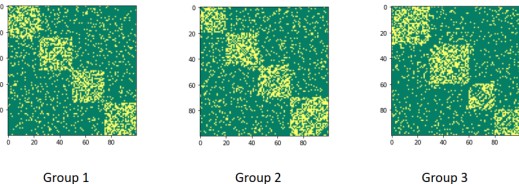

Figure 2: The adjacency matrices correspond to a single layer for each group. $G^1$ is presented on the right, $G^2$ in the middle, and $G^3$ on the left. The intra-block density is set at $\pi_{\text{intra}} = 0.5$, and for better visualisation, the inter-block connectivity probability is set at $\pi_{\text{inter}} = 0.1$, instead of $\pi_{\text{inter}} = 0.3$ as really experimented.

The MSBM and the K-means results are shown in Table 1. The MSBM successfully finds the appropriate layer-to-group assignments. In contrast, the K-means approach fails to perform this task. The application of multiplex clustering algorithm results in an average clustering of the vertices. Unlike MSBM, there is no distinction between blocks of vertices for each group of layers, which explains the better NMI/AMI results of the MSBM over the multiplex SBM, as illustrated in Table 1. The mean errors between the predicted and generated parameters is $0.04$, which proves the enhancement of parameter recovering of MSBM regarding multiplex SBM make error of $0.67$.

| Metrics \ Algorithms | Layer-to-Group | | Vertex-to-Block | | | |
|---|---|---|---|---|---|---|
| | MSBM | K-means | MSBM | SBM | GLouvain | GFSC |
| NMI | 100 | 52.03 | 100 | 61.40 | 77.44 | 73.80 |
| AMI | 100 | 49.40 | 100 | 58.74 | 75.74 | 72.65 |

Table 1: The NMI and AMI performances on MSBM, K-means, SBM, Glouvain and GFSC in varibility on block size synthetic datasets.

## 6 CONCLUSION

Throughout this paper, we have introduced the Mixture Stochastic Block Model (MSBM) for multi-group community detection in multiplex graphs. The MSBM serves to infer the existing groups that share a similar community structure. Therefore, for each identified group, a distinct SBM is derived to ascertain the community structure of each vertex. We have devised an Expectation-Maximization (EM) framework for the estimation of layer-to-group assignment variables, followed by a Variational EM technique for estimating vertex-to-block assignments. A novel centroid methodology has been proposed to initialize both layer-to-group and vertex-to-block variables, enhancing the model's convergence.

This model has been formulated with the intent of refining the estimation of the generating model underlying multiplex graphs. It significantly contributes to a better comprehension of community structures within multiplex graphs characterized by multi groups of community memberships. While the current presentation exclusively addresses unweighted graphs, there exist potential extensions encompassing degree correction and the incorporation of weights through alternative probability distributions such as Gaussian or Poisson distributions. Such extensions would undoubtedly enrich the model's applicability in capturing the intricacies of diverse real-world scenarios.

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

## A SUPPLEMENTARY EXPERIMENTS

### A.1 SYNTHETIC DATA: VARIABILITY IN BLOCK DISTRIBUTION

In this experiment, the dataset consists of 3 groups of 10 layers. Each graph is composed of 100 vertices distributed over four blocks of the same size ($\{25, 25, 25, 25\}$) and the same $\pi_{intra} = 0.5$. The layers' groups difference is characterized by the probability of having an edge between the blocks in which $\pi_{inter}^{G^1} = 0.1$, $\pi_{inter}^{G^2} = 0.3$ and $\pi_{inter}^{G^3} = 0.5$. This difference enables us to test the algorithm's ability to find groups with different $\mathbf{\Pi}$ distributions. One can notice that the third group is characterized by $\pi_{inter} = \pi_{intra}$, which corresponds to a random graph without communities.

The MSBM accurately identifies the clusters in both vertex and layers clustering with an error of estimated parameters equal to $mean_e = 0.04$, such that that for the random layers without communities, the algorithm does not find $4$ clusters but considers only "1 community". The K-means is still bad at retrieving the layers group, and the multiplex algorithms only manages to find an average effect for vertex-to-block assignment with higher estimation error compared to MSBM.

| Algorithms Metrics | Layer-to-Group | | Vertex-to-Block | | | |
|---|---|---|---|---|---|---|
| | MSBM | K-means | MSBM | SBM | GLouvain | GFSC |
| NMI | 100 | 15.50 | 100 | 55.54 | 66.66 | 53.55 |
| AMI | 100 | 10.10 | 100 | 55.03 | 66.66 | 53.02 |

Table 2: The NMI and AMI performance for MSBM, K-means, SBM, Glouvain and GFSC in variability on block size synthetic datasets.

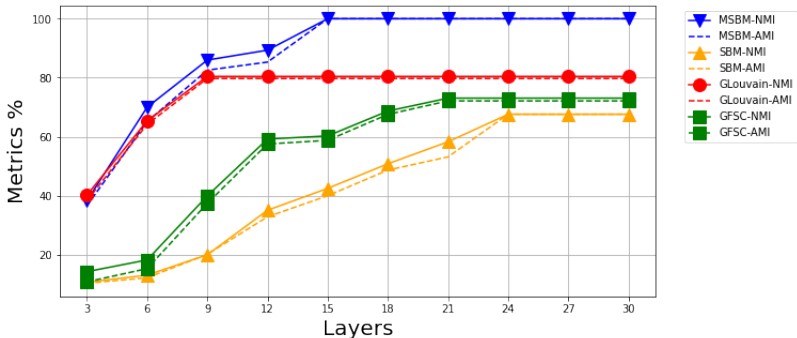

Figure 3: The perfomance of MSBM, SBM, Glouvain ad GFSC to find the clusters of vertices regarding the number of layer. The NMI and AMI were used are metrics of performance

## A.2 SYNTHETIC DATA: VARIABILITY IN NUMBER OF LAYER

In this experiment, we fix the number of vertices, the number of groups and the blocks distribution for each group, and we variate the number of layers. As the experiment for the variability in block size, we set three groups with equitable number of layers and different block division for each, such that $\pi_{intra} = 0.5$ and $\pi_{inter} = 0.3$. The blocks within the groups are divided into four blocks such that $G^1 = \{25, 25, 25, 25\}$, $G^2 = \{20, 25, 25, 30\}$ and $G^3 = \{30, 30, 20, 20\}$, where $G^i$ is the $i^{th}$ group.

The result of the following experiments is shown in the figure 3. We can see that the performance of the MSBM to retrieve the optimal blocks for each layer augment when the number of the layer augment, comparing to the other methods. It can be explainable by the law of large number that describes the convergence in probability to the expected value as the number of samples increases, which are the layers in our case. As the layers of the multiplex graph augments, the time of computation augments linearly in this case because number of parameters that scale linearly with the number of layer for fixed number of groups and blocks, differently from the other tested algorithms that does not scale well with large datasets. Additionally, thanks to our initialization that help to converge to good local minimum, up to more than 10 time faster than the random initialization.

## A.3 SYNTHETIC DATA: VARIABILITY IN NUMBER OF VERTICES

In this experiment, we fix the number of layer, the number of groups and the block distribution of each group, and then we variate the size of graph by variating its number of vertices. We set three groups with equilibrium number of layers and different block division for each one, as the previous experiments.

The obtained result is represented in the figure 4. We can notice that the MSBM scale to large dataset with thousand of nodes. The time complexity variate regarding size of graph and the structure of the block. The computation time of the MSBM model is quiet small, (less than 20 second for the multiplex graph with 900 vertices and 30 layers). It is explainable by the good initialization that it may be returned from the centroid model. We recall that the EM technique is sensitive to initialization, such that a random initialization may take longer time to stuck a fixed point.

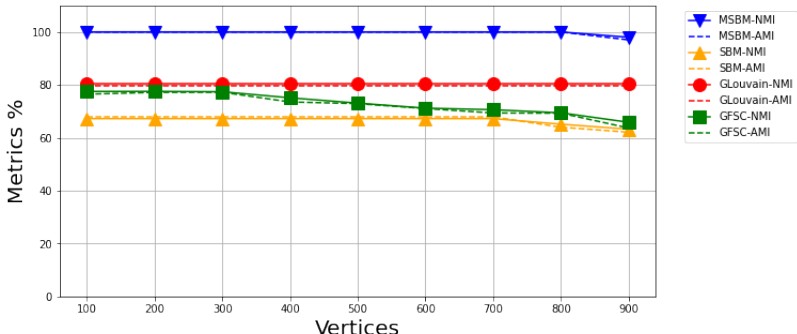

Figure 4: The perfomance of MSBM, SBM, Glouvain ad GFSC to find the clusters of vertices regarding the number vertices in multiplex graph. The NMI and AMI were used are metrics of performance

## A.4 REAL WORLD DATA

In order to evaluate the practical effectiveness of the MSBM model within a real-world context, we generate a composite dataset by joining multiple datasets, each inherently possessing its own ground truth labels. To do so, we employ the *BBC*[1] and *BBCSport*[2] datasets. The *BBC* dataset consists of $2,225$ articles divided into five different categories. It contains nine views, where each view represents the sequence {articles, words}. The datasets are summarized in Table 3. To adapt this dataset to our framework, we need to reconstruct a classical article-article graph. We did so by interconnecting articles that share common words. The arising multiplex graph has integer weights, which represent shared words among articles. For the purpose of applying MSBM based on the Bernoulli distribution, we opt to select edges with weights exceeding the mean weight across all the graphs. However, it may be more suitable to model the MSBM by Poisson distribution, which is more convenient for integer weights. Poisson MSBM distribution is out of the scope of this paper.

| dataset | individuals | views | clusters |
|---------|-------------|-------|----------|
| BBC | 2,225 | 9 | 5 |
| BBCSport | 737 | 9 | 5 |

Table 3: Summary of real datasets

We then test the ability of the MSBM to distinguish between those nine layers and nine randomly generated layers with $\pi_{\text{inter}} = 0.3$, $\pi_{\text{intra}} = 0.5$ uniformly distributed in $4$ clusters. From the results shown in Table 4, the MSBM perfectly distinguishes the two groups. One can see that our method is better than K-mean, SBM, GLouvain and GFSC is both identifying group and clustering. Improvement of clustering can be done also by relaxing the assumptions regarding the the homogeneous distribution within each block, e.g. by considering degree-corrected variants of the SBM Qin & Rohe (2013). This method allows for varying distributions among vertices within the same block, thus enhancing the model's fidelity to real-world complexities. The degree-corrected MSBM is however out of the scope of this paper and left for potential future work, especially that our work provides improvements as compared to existing work in this area as shown above.

In a second step, we tested the ability of MSBM to distinguish between two groups of the same nature but different organizations, the *BBCSport* and the *BBC* datasets. As shown in Table 5, the results are again very good and less good for the vertex-to-block assignment. Our method improves the performance in terms of detecting groups and clustering as compared to K-mean, SBM, GLouvain and GFSC schemes.

---

[1]http://mlTheg.ucd.ie/datasets/segment.html
[2]http://mlg.ucd.ie/datasets/segment.html

| | Layer-to-Group | | Vertex-to-Block | | | |
|---|---|---|---|---|---|---|
| Algorithm | MSBM | K-means | MSBM | SBM | GLouvain | GFSC |
| NMI | 100 | 67.5 | 49.5 | 35.6 | 43.5 | 41.2 |
| AMI | 100 | 59.4 | 48.3 | 33.5 | 41.3 | 40.1 |

Table 4: The NMI and AMI performance of MSBM, K-means, SBM, GLouvain and GFSC in BBC dataset group and randomly generated layers group.

| | Layer-to-Group | | Vertex-to-Block | | | |
|---|---|---|---|---|---|---|
| Algorithm | MSBM | K-means | MSBM | SBM | GLouvain | GFSC |
| NMI | 100 | 50.40 | 52.14 | 30.47 | 40.68 | 33.54 |
| AMI | 100 | 47.40 | 50.29 | 28.47 | 40.10 | 30.39 |

Table 5: The NMI and AMI performances of MSBM, K-means, SBM, GLouvain and GFSC in *BBC* and *BBCSport* datasets.

# B  IDENTIFIABILITY

The identifiability of the parameters for uni layer Bernoulli SBM has been proved in Celisse et al. (2012). The proof has been extended to a multiplex graph for pillar division Barbillon et al. (2017). We extend this analysis for multiplex SBM with multi groups.

**Theorem 1.** *Let assume that there is $K$ groups and every group has the same number of blocks $Q^k = Q^{k'} = Q \ \forall k, k' \in \{1, ..., Q\}^2$. Assume for any $q \in \{1, ..., Q\}, k \in \{1, ..., K\}, \alpha_q^k > 0, \beta^k > 0$. Let $\mathbf{\Pi} \in ]0, 1[^{K*Q \times K*Q}$ diagonal block that contains matrices $\Pi^k$ at diagonal as follow:*

$$\begin{bmatrix} \mathbf{\Pi^1} & ... & 0 \\ \vdots & ... & \vdots \\ 0 & ... & \mathbf{\Pi^K} \end{bmatrix}$$

*Let also $\boldsymbol{\alpha}$ be a $K*Q \times K*Q$ matrix, which is the diagonilization of $[\alpha_1^1, ...\alpha_Q^1, ...\alpha_Q^K]$ vector, and $\boldsymbol{\beta}$ be a $K*Q \times K*Q$ matrix, which is the diagonilization of $[\beta^1, ...\beta^1, \beta^2...\beta^K]$ vector, where $\beta^i$ is repeated $Q$ times, $\forall i \in \{1, ..., K\}$. Assume that the elements of $\boldsymbol{r} = \mathbf{\Pi}.\boldsymbol{\alpha}.\boldsymbol{\beta}$ are distinct. Then the MSBM parameters are identifiable*

———

*Proof.* We extend the proof from Celisse et al. (2012) to the MSBM model as follows. . For any group $k$, $r_{q,k}$ is the probability for a giving member from block $q$ in group $k$ to have a connection with another in the same group $r_{q,k} = \sum_{l=1}^{Q} \beta_k \pi_{ql}^k \alpha_l^k$. Let $\mathbf{R}$ be $Q*K$ square matrix such that $R_{i,q,k} = (r_{q,k})^i$ for $i \in 0, ..., Q*k - 1$. $R$ is a Vandermonde matrix that is invertible by assumptions.

Therefore, for any $i = 0, ..., (2Q - 1)*K$, let set

$$\mu_i = \sum_{q,k} \alpha_{q,k}(r_{q,k})^i \tag{24}$$

and $M$ is a $k(Q + 1) \times KQ$ matrix such that

$$M_{ij} = \mu_{i+j} \tag{25}$$

For any $i = 0, ..., Q*k$, we define the Q*K square matrix $M^i$ by removing line $i$ from the matrix. In hence,

$$M^Q = R\boldsymbol{\alpha}R^T \tag{26}$$

where $\boldsymbol{\alpha}$ is $Q*K$ matrix as defined previously, where all $\alpha_q^k \neq 0$. Because $R$ being invertible, then $det(M) > 0$. Let us define

$$B(X, \theta) = \sum_{i=0}^{Q \times K} (-1)^{i+Q*K} det(M^i(\theta))X^i \tag{27}$$

$B$ is of degree $Q \times K$. For $V^i(\theta) = (1, r_i(\theta), ..., (r_i(\theta))^Q)$, then

$$B(r_i(\theta), \theta) = det(M(\theta), V_i(\theta)) \tag{28}$$

The column of $M$ are linearly combinations of $V_i$, then $B(r_i(\theta), \theta) = 0$ for any $i$. It means that $B$ can be factorized as follow:

$$B(x, \theta) = det(M^{Q \times K}) \prod_{i=0}^{KQ-1} (x - r_i(\theta)) \tag{29}$$

Let assume the $\boldsymbol{\theta} = (\boldsymbol{\Pi}, \boldsymbol{\alpha}, \boldsymbol{\beta})$ and $\boldsymbol{\theta'} = (\boldsymbol{\Pi'}, \boldsymbol{\alpha'}, \boldsymbol{\beta'})$ are two sets of parameters such that for any multiplex $\mathcal{G}$ graph with multi-group model, $\mathcal{L}(\mathcal{G}; \boldsymbol{\theta}) = \mathcal{L}(\mathcal{G}; \boldsymbol{\theta'})$. Therefore, we get, $\mu_i(\boldsymbol{\theta}) = \mu_i(\boldsymbol{\theta'})$, that means that $M^i(\theta) = M^i(\theta')$ for any $i$. The $B(; \theta) = B(; \theta')$ because it dependents on the determinant of $M$, which leads to $r_i(\theta) = r_i(\theta')$. Ths $R(\theta) = R(\theta')$, and

$$\boldsymbol{\alpha}(\theta) = (R(\theta)^T)^{-1} M^{Q,K} R(\theta) = \boldsymbol{\alpha}(\theta') \tag{30}$$

Therefore $\boldsymbol{\alpha} = \boldsymbol{\alpha'}$. The same steps can be applied to proof the identifiability of $\boldsymbol{\beta}$ where the matrix diagonal $\boldsymbol{\alpha}$ is replaced by diagonal matrix of $\boldsymbol{\beta}$ where every $\beta_k, \forall k \in \{1, ..., K\}$ will be repeated $Q$ times before set $\beta_{k+1}$. It leads to a matrix with the same dimension $Q \times K$.

For $\boldsymbol{\Pi}$, let's define

$$U_{ij} = R(\boldsymbol{\theta})\boldsymbol{\beta}(\boldsymbol{\theta})\boldsymbol{\alpha}(\boldsymbol{\theta})\boldsymbol{\Pi}\boldsymbol{\alpha}(\boldsymbol{\theta})\boldsymbol{\beta}(\boldsymbol{\theta})(R(\boldsymbol{\theta}))^T$$

From previously, $R(\theta) = R(\theta')$, $\boldsymbol{\alpha}(\theta) = \boldsymbol{\alpha}(\theta')$ and $\boldsymbol{\beta}(\theta) = \boldsymbol{\beta}(\theta')$ then

$$U(\boldsymbol{\theta}) = U(\boldsymbol{\theta'}) \rightarrow \boldsymbol{\Pi} = \boldsymbol{\Pi'} \tag{31}$$

$\square$

## C CONSISTENCY OF MAXIMUM LIKELIHOOD

The asymptotic consistency of the maximum likelihood estimator of Bernoulli uni layer SBM has been studied in Celisse et al. (2012). The proof of the consistency of MSBM is straightforward from the proof of uni-layer SBM. Let's assume that the following assumptions hold:

**Assumption 1.** *For every $q \neq q'$, there exists $w \in \{1, ..., Q^k\}$ such that $\pi_{qw}^k \neq \pi_{q'w}^k$, or $\pi_{wq}^k \neq \pi_{wq'}^k$*

**Assumption 2.** *There exists $\zeta > 0$ such that $\forall (q, w) \in \{1, ..., Q^k\}$, $\pi_{qw}^k \in ]0, 1[ \rightarrow \pi_{qw} \in [\zeta, 1 - \zeta]$*

**Assumption 3.** *There exists $\gamma \in 1/Q^k$ such that $\forall q \in \{1, ..., Q^k\}$, $\alpha_q^k \in ]0, 1[ \rightarrow \alpha_q \in [\gamma, 1 - \gamma]$*

**Assumption 4.** *There exists $\xi \in 1/K$ such that $\forall k \in \{1, ..., K\}$, $\beta^k \in ]0, 1[ \rightarrow \beta^k \in [\xi, 1 - \xi]$*

**Theorem 2.** *Let $(\Theta, d)$ and $(\Psi, d')$ denote metric spaces and let $M_n : \Theta \times \Psi \rightarrow \mathcal{R}$ be a random function and $M : \Theta \rightarrow \mathcal{R}$ a deterministic function such that for every $\epsilon > 0$*

$$sup_{d(\theta, \theta_0)} M(\theta) < M(\theta_0) \tag{32}$$

$$sup_{(\theta, \psi) \in \Theta \times \Psi} |M_n(\theta, \psi) - M(\theta)| := ||M_n - M||_{\Theta \times \Psi} \rightarrow 0 \tag{33}$$

*and $(\hat{\theta}\hat{\psi}) = \underset{\theta, \psi}{argmax} \, M_n(\theta, \psi)$, then*

$$d(\hat{\theta}, \theta_0) \rightarrow 0 \tag{34}$$

The proof can be performed by the same steps by taking

$$M_n(\pi, \alpha, \beta) = \frac{1}{N(N-1)L * K}$$

$$\sum_{l=1}^{L} log\Big(\sum_k \sum_{z_{[n]}} \beta_k \prod_{i \neq j} \textbf{Bernoulli}(\pi_{z_i^k, z_j^k}^k) \prod_i \alpha_{z_i^k}^k\Big) \tag{35}$$

$$M(\pi) = \max_{a_{i,j} \in \mathcal{A}} \sum_k \sum_{q,w} \beta^{*k} \alpha_q^{*k} \alpha_w^{*k}$$

$$\sum_{q', w'} a_{qq'}^{*k} a_{ww'}^{*k} [\pi_{q,w}^{*k} log(\pi_{q',w'}^{*k}) + (1 - \pi_{q,w}^{*k}) log(1 - \pi_{q',w'}^{*k})] \tag{36}$$

where **Bernoulli**$(\pi)$ is the Bernoulli distribution with paramter $\pi$, and $\beta^*, \alpha^* and \pi^*$ denotes the true parameters respectively, with

$$\mathcal{A} = \{(a_{ij}^k)_{1 \leq q, w \leq Q^k}, a_{qw}^k \geq 0, \sum_w a_{qw}^k = 1\} \tag{37}$$

## D  OPTIMIZATION OF INITIALIZATION MODEL

The initialization model described in Section 4.3 in Equation (21) involves multiple variables, making it challenging to optimize them simultaneously. Therefore, we adopt an iterative technique where each variable will be optimized while the others are held fixed.

### D.1  OPTIMIZING **Y**, WHILE **U** AND **F** ARE FIXED

The model can be represented as follows:

$$\min_{\mathbf{Y}} \sum_{l=1}^{L} \sum_{k=1}^{K} y_{lk} ||\mathcal{U}^k - \mathcal{A}^l||_F^2$$
$$s.t \ y_{lk} \in \{0,1\}, \sum_{k=1}^{K} y_{lk} = 1 \tag{38}$$

By relaxing the constraint $y_{lk} \in \{0,1\}$ to $y_{lk} \in [0,1]$, the model becomes linear, facilitating the application of analytical solutions that satisfy the Karush-Kuhn-Tucker (KKT) conditions using the Lagrange technique. The analytical solution is expressed as follows:

$$y_{lk} = \frac{||\mathcal{U}^k - \mathcal{A}^l||_F^2}{\sum_{k'=1}^{K} ||\mathcal{U}^{k'} - \mathcal{A}^l||_F^2} \tag{39}$$

The determination of the group to which the layer $l$ will be assigned is carried out as follows:

$$y_{lk} = \underset{k}{\operatorname{argmax}} \ y_{lk} \tag{40}$$

### D.2  OPTIMIZING $\mathcal{U}$, WHILE $Y$ AND **F** ARE FIXED

Firstly, the optimization of centroids $\mathcal{U}^k$ is performed independently, and according to Wang et al. (2020), the objective function $Tr(\mathbf{F^k}^T \mathbf{L}_{U^k} \mathbf{F^k})$ can be expressed as $\sum i,j ||\mathbf{f}_i - \mathbf{f}j||_2^2 ui,j$. Therefore, the optimization for each centroid can be formulated as follows:

$$\min_{\mathcal{U}^k} \sum_{l \in L^k} \sum_{i,j} ||u_{ij} - A_{ij}||_F^2 - \lambda \sum_{i,j} ||\mathbf{f}_i - \mathbf{f}_j||_2^2 u_{i,j}$$
$$s.t \ u_{ij} \geq 1, \mathbf{1}^T.\mathbf{u}_i = 1 \tag{41}$$

$L_k$ represent the set of layers for which $y_{lk}$ equals one. We denote $||\mathbf{f}_i - \mathbf{f}j||_2^2$ as $dij$. Due to the independence of optimization for each vertex vector $\mathbf{u}_i$ from the others, the optimization of $\mathcal{U}^k$ can be formulated as follows:

$$\min_{\mathbf{u}_i^k} \sum_{l \in L^k} ||\mathbf{u}_i - \frac{\lambda}{2|L^k|}\mathbf{d}_i||_F^2$$
$$s.t \ \forall i,j \ u_{ij} \geq 1, \mathbf{1}^T.\mathbf{u}_i = 1 \tag{42}$$

The model mentioned above is quadratic with linear constraints, indicating that it is convex. It can resolved using the augmented Lagrange multipliers. Otherwis, any quadratic solver can efficiently resolve this problem.

### D.3 OPTIMIZING $\mathbf{F}^k$ WHILE $\mathbf{U}$ AND $\mathbf{Y}$ ARE FIXED

The optimization of each $\mathbf{F}^k$ for every group is performed independently of the remaining groups. For a given group, the model can be formulated as follows:

$$\min_{\mathbf{F}^k} Tr(\mathbf{F}^{k^T} L_{\mathcal{U}^k} \mathbf{F}^k)$$

$$s.t \ \ \mathbf{F}^{k^T}.\mathbf{F}^k = \mathbf{I}$$

(43)

The optimal $\mathbf{F}^k$ can be obtained by extracting $Q^k$ eigenvectors of the Laplacian matrix $L_{\mathcal{U}^k}$, which are associated with the smallest $Q^k$ eigenvalues. It is important to note that $Q^k$ denotes the number of communities within group $k$.

---

**Algorithm 2** Multi Centroids algorithm fo initialization

---

**Input** $\mathcal{G}, K, \mathbf{Q} = [Q^1, ..., Q^K]$ **Output**: $\mathbf{U}, \mathbf{Y}$
  Initialize $\mathbf{U} = \{\mathcal{U}^1, \mathcal{U}^2, ..., \mathcal{U}^K\}$
  **while** Iteration $<$ Iteration max $||$ Not Converge  **do**
    optimize $y_{lk}$ with 40
    optimize $\mathbf{u}_i$ with 42
    compute $\mathbf{F}^k$ from the eigen vector associated to $Q^k$ smallest eigen value of $L_{\mathcal{U}^k}$
  **end while**

---

