# OpenReview forum: "Mixture Stochastic Block Model for Multi-Group Community Detection in Multiplex Graphs"
_ICLR.cc/2024/Conference — Submitted to ICLR 2024_

### Official Review · Reviewer_MACM · 2023-10-23

**Soundness:** 2 fair
**Presentation:** 3 good
**Contribution:** 2 fair
**Rating:** 5
**Confidence:** 5

**Summary:**

The paper introduces the "Mixture Stochastic Block Model," a proposed method for grouping similar layers based on shared community structures in multiplex graphs. Instead of focusing on individual layer clusters, the approach aims to identify community structures across different layers. The model is carefully defined, and an iterative technique is utilized for computing the inference. The paper includes analyses of identifiability and consistency, providing theoretical support for the proposed method. The performance of the approach is evaluated using synthetic graphs and real-world datasets, effectively showcasing its ability to identify consistent community structures in diverse multiplex graphs.

**Strengths:**

S1. The paper demonstrates good writing and logical organization, making it reader-friendly.

S2. The authors provide codes of the proposed method, indicating the reproducibility of the study.

S3. Including identifiability and consistency analyses distinguishes the proposed method from current deep learning approaches, providing theoretical underpinnings.

**Weaknesses:**

W1. The motivation needs to be strengthened. It is unclear why research on multi-group community detection in multiplex graphs is important. Although some studies on multiplex graphs exist, the scarcity of research on multi-group community detection may be due to the difficulty of the topic or its limited occurrence in the real world. Please clarify the reasons behind the lack of research or provide examples from the real world to support the importance of this study.

W2. The paper lacks novelty. While the authors mention Stanley et al. (2015) as existing literature on multi-group community in multiplex graphs, the paper only highlights the difference between this work and Stanley et al. (2015) in terms of the learning of layer-to-group assignments, with the former using the EM algorithm instead of k-means. What are the challenges in replacing k-means with the EM algorithm? Are there any differences in the modeling assumptions between this paper and Stanley et al. (2015)?

W3. The experimental section is weak. Firstly, the paper does not compare the proposed model, which incorporates layer-to-group assignments, with other models that handle multiplex graphs but do not consider layer-to-group assignments [1,2,3,4]. This comparison is necessary to highlight the importance of layer-to-group assignments. Secondly, the datasets used in the experiments are relatively small, while real-world data often exhibit large-scale characteristics. What is the time complexity of the proposed algorithm? Can it handle large-scale data?

W4. The references are incomplete. For example, [1,2,3,4] are all methods for multilayer graphs, but the authors did not cite them. Although [4] focuses on dynamic networks, its modeling approach for generating multilayer networks is similar to the approach in this paper. I recommend that the authors cite and analyze this reference.

References:
[1] Han Q, Xu K, Airoldi E. Consistent estimation of dynamic and multi-layer block models[C]//International Conference on Machine Learning. PMLR, 2015: 1511-1520.
[2] De Bacco C, Power E A, Larremore D B, et al. Community detection, link prediction, and layer interdependence in multilayer networks[J]. Physical Review E, 2017, 95(4): 042317.
[3] Paul S, Chen Y. Consistent community detection in multi-relational data through restricted multi-layer stochastic blockmodel[J]. 2016.
[4] Corneli M, Latouche P, Rossi F. Exact ICL maximization in a non-stationary temporal extension of the stochastic block model for dynamic networks[J]. Neurocomputing, 2016, 192: 81-91.

**Questions:**

Q1. What are the reasons behind the scarcity of research on multi-group community detection in multiplex graphs? Are there any challenges or limitations in studying this topic?

Q2. In terms of novelty, are there any differences in the modeling assumptions between this paper and Stanley et al. (2015)? Please clarify.

Q3. What are the reasons the EM algorithm performs better than Kmeans?

Q4. Can you compare the proposed model with other models that handle multiplex graphs but do not consider layer-to-group assignments? This comparison would highlight the importance of layer-to-group assignments.

Q5. What is the time complexity of the proposed algorithm? Can it handle large-scale data?

---

> ### Author Response · Authors · 2023-11-22
>
> We appreciate your valuable review feedback. All your comments have been thoroughly considered, and we have implemented the necessary revisions to the paper in accordance with your suggestions. To clarify the motivation behind our model, consider a scenario involving multi-modal data, such as that used in a recommendation system. Users' similarity may be assessed based on information from multiple sources that are not directly comparable. Therefore, we represent this data as a multiplex graph. Now, imagine the task of making recommendations for a user based on the communities to which they belong. One might consider constructing clusters independently for each layer or constructing clusters that take into account all layers simultaneously. The former approach is time-consuming and may not leverage information from other layers effectively, while the latter may yield a consensus division that is not optimal for all layers. Our proposed method strikes a balance between these two models. We first identify a group of layers, and then we cluster each group. This approach allows us to obtain more refined and accurate recommendations for items.
>
> In Stanley (2015), the utilization of K-means on the parameters introduces a potential limitation as K-means assumes well-behaved spherical topological spaces, which might lead to erroneous clusters. In contrast, our model presents a coherent and well-structured pipeline that relies solely on probability distributions. Our optimization process is centered around likelihood optimization. While our assumptions align with those in the stochastic block community, we distinguish ourselves through our logical approach, reliance on probability distributions
>
> As best of our knowledge, there is a lack of open-source labeled datasets suitable for this application. We aim to publish a new dataset from our field in the future. We augment our experimentation, particularly focusing on demonstrating the model's performance with large graphs. Thanks to our initilalization model that helps to get fast convergence of MSBM.
>
> Q1:This is indeed a novel and crucial question, prompted by its significance in our industrial application. The scarcity of research in this area may be attributed, in part, to the absence of open-source datasets that effectively capture the essence of this question. Our modelization approach is versatile, applicable to various multimodal or multiview datasets, aligning with current trends in machine learning clustering fields. We believe that this modelization offers a means to achieve a finer understanding of datasets, enabling the creation of models tailored to the inherent nature of similarities within the data
>
> Q2: While our modelization may share similarities with Stanley et al. (2015), we distinguish ourselves through a robust mathematical formulation that ensures asymptotic convergence to the best division. Notably, our contribution extends to offering a novel initialization method, a departure from random initialization, which significantly enhances the speed of convergence. Additionally, we contribute theoretical proofs establishing the identifiability and consistency of our model, providing a solid foundation for its reliability. Moreover, our model, with the proposed optimization techniques, easily extends to accommodate other modelizations that consider degree correction and various weighted distributions. This flexibility makes our approach versatile and applicable to a broader range of scenarios, showcasing its adaptability and potential for addressing diverse modeling challenges.
>
> Q3: Indeed, the K-means algorithm has found widespread use in various fields. However, it comes with inherent limitations, most notably its native assumption that clusters have a spherical shape. This assumption does not always align with the actual structure of the data, presenting a significant drawback. In contrast, the EM (Expectation-Maximization) algorithm does not impose any assumptions about the shape of clusters. This flexibility allows the EM algorithm to model complex distributions, making it more adaptable to the diverse structures encountered in real-world data. The EM algorithm generally converges to a good local optimum, and its versatility extends to incorporating any available prior knowledge, adding a layer of flexibility and customization to its application.
>
> Q4: We add Generalized Louvain and Graph Fusion Spectral Clusteting models algorithms to compare with, such that those algorithm are not assuming the existence of multigroup clustering as you proposed.
>
> Q5: The EM and VEM algorithms exhibit sensitivity to initialization. Thank to our spectral initialization that serves as a valuable aid, providing the model with a helpful initialization of clusters that accelerates the convergence process. We conducted new experiments in which we variate the number of layers and the number of vertices. Thank you for bringing this to our attention.

---

> > ### Comment · Reviewer_YZJF · 2023-11-22
> >
> > I thank the authors for their responses, which I will consider carefully during the reviewer discussion period to decide if I should change my score.

---

> > ### Comment · Reviewer_MACM · 2023-11-23
> >
> > Thank you for your response. It has addressed some of my concerns. However, the experiments still are insufficient, and the explanation of time complexity could be clearer.
> >
> > Considering these, I updated my score to 5.

---

### Official Review · Reviewer_VvaF · 2023-10-30

**Soundness:** 2 fair
**Presentation:** 1 poor
**Contribution:** 2 fair
**Rating:** 3
**Confidence:** 2

**Summary:**

This paper proposes a mixture stochastic block model (MSBM) for community detection in multiplex graphs. The method is based on the estimation maximization technique according to the underlying MSBM structure. The experimental results on synthetic and real datasets verify the effectiveness of this method.

**Strengths:**

A new random graph model for multiplex graphs and a new approach for multiplex graph clustering are proposed.

**Weaknesses:**

To be honest, I cannot continue reading up to Section 3.1. The model definition is quite hard to follow. Due to the pressed review time, I do not have enough time to understand it. The authors defined a large number of parameters with superscripts and subscripts for expression accuracy. However, the basic architecture of the model is not very clearly for me. To my understand, the underlying layers are proposed for the simulation of multiplexity of a graph. Each layer is partitioned into $K$ groups and each group is partitioned further into blocks (a bit weird, probably I have a misunderstanding here). The probability of presence of edges depends on the vertex-to-block assignments and the probability matrices for inter and intra blocks, just like the traditional SBM. But how do we understand that a layer is generated from a group? A layer has several groups, so I wounder how to generate this layer from a single group and how to understand vector $\beta$ and $y_{lk}$. A simple example would be much helpful, but there isn't. Since the basic architecture is not clear, I cannot evaluate the subsequent work.

**Questions:**

(1) In Section 3.1, what do you mean when you say "let's consider a partition of layer $L$ into $K$ groups and assume that group $k$ comprises $Q^k$ blocks"? What is the relationship between group and block in your definition? What's the purpose of them?

(2) Is there a simple example to help understand the notations in Section 3.1?

---

> ### Author Response · Authors · 2023-11-22
>
> Thank you for your review and we feel sorry that you found our paper difficult to follow. The new version of the paper has been written to be more readable, and we trust you will find it more clear. To better comprehend our model, consider a multilayer graph where nodes are shared across layers, and each layer possesses its distinct set of edges. This multiplex graph serves as a representation of multiview data, such that information may come from different perspectives of similarity. Our goal is to identify groups of layers that share similar characteristics, a task we refer to as group clustering. Subsequently, we aim to cluster the nodes within each group. Here, a group denotes a collection of layers or graphs, and a block signifies a set of nodes. Therefore, we work with two types of assignment variables: layer-to-group for identifying the group/cluster to which each layer belongs and vertex-to-block for assigning each node within a group to a specific block/cluster.
>
> The concept of our model is relatively novel, and its applications hold significant promise in addressing real-world problems. The variable $y_{lk}$ signifies that layer $l$ belongs to group $k$, and $\beta$ represents a vector of probabilities indicating the likelihood of a layer being generated from group $k$.
>
> Q1: As mentioned earlier, our multiplex graph comprises $|L|$ layers, and our objective is to identify $K$ groups where each group is composed of similar layers. Within each group k, our aim is to cluster its vertices into $Q^k$ clusters. The superscript $k$ is utilized to explicitly convey that each group may have its distinct number of clusters or blocks. Therefore, we apply joint clustering of layers, and clustering of vertices within each group of layers.
>
> Q2: Thank you for your suggestion. It's a relevant idea to include an example that explains the model. We Add now a figure that illustrates the application which will likely enhance the clarity and understanding of your work.

---

### Official Review · Reviewer_YZJF · 2023-10-30

**Soundness:** 3 good
**Presentation:** 3 good
**Contribution:** 2 fair
**Rating:** 3
**Confidence:** 3

**Summary:**

The aim of the paper is to compute shared clusters/communities across layers of a multiplex graph that exhibit similar community structures, and separate clusters/communities for layers exhibiting different community structures. It is clearly motivated why having (potentially) different groups of communities across layers may be desirable and how it is not sufficiently captured by existing work on Multiplex Community Detection.

The method consists of two parts: Grouping layers with expectation-maximization technique and assigning vertices to communities across grouped layers with a variational estimation-maximization technique. Compared to existing variational estimation-maximization technique for community detection, this reduces the complexity of parameters that need to be fitted which is desirable for high-dimensional datasets.

**Strengths:**

* Good motivation and promising underlying idea.
* Sound theoretical foundation

**Weaknesses:**

None of the presented state-of-the-art methods from the related work sections are compared against. Only two naive approaches, K-means and the native SBM, are used as competing methods -- even though their drawbacks were made abundantly clear in the paper. Here, some validation on how well the communities in recovered multiplex SBM describes/generate the observed data in comparison to other community detection methods would be useful instead of only considering the two individual steps of the presented approach independently. Such a comparison might also contain synthetic SBMs with only shared groups as assumed by many existing techniques as a baseline and their failure to capture non-shared groups (highlighting the improvement of the new MSBM method).

The description of task 5.1 is confusing as it is initially refereed to as the vertex-to-block assignment task, but within the same paragraph it is described how the NMI is used to evaluate the layer-to-group association. I am assuming you compared your method which takes all layers into account to 30 independently fitted SBMs which does not seem like a fair comparison; however, this might just be a misunderstanding due to the confusing structure of 5.1 and 5.2. and missing details regarding the actual tasks that are performed.

For the task 5.2 multiple multiplex SBMs with different parametrizations should be considered to sample synthetic data for the graph clustering, like e.g. varying block or group sizes and varying blocks across layers. It is also not clear to me which different graphs clusters the task 5.2 tries to recover as only a single parametrization of an SBM is given. I am guessing that the clustering is not done on the level of Multiplex graphs, but on the level of their individual layers as graphs which then leads to a very small dataset of just 30 graphs with 3 very distinct inter-block probabilities that might be considered trivial to recover in monolayer graphs. It is not clear to see how recovering these graphs is an improvement to the state of this art. To highlight potential benefits various other graph clustering methods should be considered. However, this might also be a misunderstanding because initially I would have expected a clustering of the actual multiplex graphs based on their MSBMs when hearing “Graph clustering”.

Furthermore, I would have liked to see the impact of larger graphs have on the computational complexity as currently only small graphs of size 100 are considered. This is especially crucial as you mentioned that other competing approaches like MLSBMs suffer from exponentially large parameter space which drawbacks should be highlighted on large graphs. For this the runtime or some similar measure should be included in the results. Additionally, the impact of varying levels of noise on recovering the clusters would also be an interesting addition to the synthetic experiments.

Regarding the real-world application, it seems as task 1 was to differentiate between real datapoints and randomly generated data points which also seems to not be a suitable benchmark. Task 2 seem to differentiate the article-article graphs from two different datasets which however makes me wonder what the two rows for each dataset in Table 4 refer to as I would expect a single group-to-layer resp. vertex-to-block classification result. Maybe you can clear this up by properly introducing how the mutual information is used in this task. I also would like to ask why the adjusted mutual information (adjusted to random chance) was not used instead.

In summary, despite the well-motivated need for methods such as this, the promising underlying idea and sound theoretical foundation, it is not clear that the presented methods actually achieve this or improves the state-of-the-art. The experimental section does not quantify the quality of the recovered (Mixture) Multiplex SBM as it only shows that it performs better in the two individual steps compared to very naïve approaches.

**Questions:**

Concrete Suggestions to improve the score:

1.	Describe the actual performed task in the experiment section and make it more clear what you refer to as graph clustering (in the context of Multiplex Graphs).

2.	Include state-of-the-art methods and actually compare the recovered Multiplex SBMs instead of only the results of individual steps of MSBM

3.	(Maybe) include a larger synthetic dataset w.r.t the number of graphs, nodes, blocks and different sets of parameters to seem less exemplary.

4.	Include more metrics than just NMI and a run time comparison


Things to improve the paper that did not impact the score:

No statement about reproducibility possible as no code was provided.

The motivation to leverage Multiplex graphs as multi relational data representation and their benefits are clear. But inherent limitations of Multiplex graphs like the requirement of actors to be identical across types of interactions/layers are not addressed or mentioned. Although being a very powerful tool, it most likely is not the solution to all complex multi-dimensional datasets. I would like the authors to very briefly address and differentiate this to Multilayer Graphs (where inter-layer edges are used to model an n-to-m mapping between actors of nodes) and, potentially, outline how the presented approach could be generalized (or not) to this data representation as part of future work.
The categorization of existing techniques into flattening, consensus and direct approaches and the placement of the new approach overall seems reasonable, however the third category of “direct approaches” seems to be very broad and might benefit from being further subdivided.

Further questions:
It is not completely clear to me why similar results could not be achieved through consensus-based approaches that support some relaxation of the consensus across all layers to only consensus across groups of layers exhibiting similar community structures and allowing a disagreement between layers of different groups akin to the approach in this paper.

What does “Layers of the same strata present their groups,…” mean? (in section 2)

---

> ### Author Response · Authors · 2023-11-22
>
> We are grateful for your insightful review feedback. All your comments have been thoroughly considered, and we have incorporated the necessary revisions into the paper according to your suggestions. We have structured the related work section to provide an overview of existing approaches in multiplex community detection. Our intention is to highlight that none of them adequately addresses the specific question we explore, thereby introducing a novel modeling approach, as detailed in the paper. The utilization of the K-means algorithm serves the purpose of identifying layers that share common clusters. In contrast, the application of the Stochastic Block Model (SBM), which assumes one blocks devision for all layers, demonstrates the model's limitation in defining distinct block distributions for each layer. We compare also the MSBM with Generalized Louvain and Graph Fusion Spectral Clustering, which are method for multiplex clustering. We use the NMI and we add the Adjusted Mutual Information (AMI) to quantify the average performance across all layers and provide a clearer assessment of the model's effectiveness
>
> In Section 5.1, we explore the impact of varying block sizes within each group and examine whether the Multiplex Stochastic Block Model (MSBM) can effectively identify the optimal clusters. Given the joint computation of group-layer and vertex-block assignments, evaluating the performance of both aspects is crucial. To elaborate, layers with different block sizes possess distinct structures, and our model aims to accurately define these differences. As mentioned earlier, our method is compared to many multiplex clustering algorithms.  We have revised the text to enhance reader-friendliness. In Section 5.2, unlike Section 5.1 where we considered varying sizes of SBM, we now explore different distributions of SBM within each group. Rather than altering all layer distributions, we opt for simplicity by fixing the intra-distribution and modifying the inter-distribution. To improve clarity, we will revise this section for enhanced readability. Furthermore, we asses the performance of the model when the number of layer and the number of node are variating, showcasing the model's performance and runtime scalability.
>
> Our model clusters the layers within the multiplex graph. If we posses a multiple multiplex graphs and we want to clusters their layers, we can use the MSBM for this application. To achieve this, we concatenate all multiplex graphs into a single comprehensive multiplex graph. The MSBM is then employed to cluster these multiplex graphs by identifying groups of layers that yield similar divisions. Each of these groups can be interpreted as a distinct multiplex graph. However, if the application is to group the similar multiplex graphs into sets of graphs, we think that MSBM is not suitable to this case.
>
> In real-world applications, the primary objective of Task 1 is to illustrate the method's capability to discern random distributions within actual datasets. We underscore its significance in scenarios where random data may be present in real applications. Concerning Task 2, we introduce new line for model performance on dataset.
>
> Q1:In our case, the graph clustering is used to mean grouping the similar layers of multiplex graphs. We make a new more clear version of the experiment set for this point.
>
> Q2: We now compare our MSBM algorithm to single SBM, Generalized Louvain and GFSC, all this comparison are shown in the experiment section.
>
> Q3: We set experiments on block size and block distribution. We add a new experiments on number of layers and the vertices size.
>
> Q4:  We now have added Adjusted Mutual Information AMI for clustering comparison.
>
> Q5: Maybe you did not pay attention, but we provide the code that we used in our experimentation.
>
> Q6: It is evident that the multiplex graph has certain limitations in relation to the general definition of multilayer graphs with edges between layers. However, as you rightly pointed out, the multiplex graph representation proves to be a powerful approach, particularly in handling complex topological spaces within multiview data structures. We believe that MSBM can be extended to encompass more general multilayer graphs. For the related section, we chose not to focus into the third category due to a lack of a comprehensive pipeline that adequately presents this category
>
> Q7: Our application could potentially be modeled using a consensus-based technique, incorporating the constraints you mentioned. However, to the best of our knowledge, there is currently no existing model capable of achieving this. Despite this, the consensus model faces a significant drawback: once the graphs are subdivided, the graph is constrained to reflect only the information division, potentially inadequately capturing the underlying graph's data structure. Additionally, our model operates as a generative model, enabling the creation of data with inherent structural information.

---

### Official Review · Reviewer_tJJH · 2023-11-01

**Soundness:** 3 good
**Presentation:** 3 good
**Contribution:** 3 good
**Rating:** 5
**Confidence:** 4

**Summary:**

The paper proposes to model the multiplex graphs using a mixture of stochastic block models. Compared to a standard stochastic block model, the use of mixture enables clustering of the graph layers into groups, each group having the same parameters. The model is optimized using Variational EM algorithm, and a spectral initialization approach is used to speed up convergence. Experiments on several synthetic data and a real world data is included.

**Strengths:**

1. The presentation and writing is clear.
2. Sufficient background is provided.
3. The idea of using spectral initializaiton to speed up convergence is very useful in practical but often left out from papers. It's nice to see an innovative approach included in this paper.
4. Theory of identifiability and consistency of MLE are provided or briefly discussed.

**Weaknesses:**

1. While the idea is interesting, the contribution seems insufficient. I would be more inclined to give a higher rating if the mixture model is presented more generally for degree corrected mixed membership models (e.g. https://arxiv.org/abs/1708.07852).
2. Most of the experiments are synthetic, and the only real world data is sort of manually constructed. It would be better if this model is application driven by more and larger real datasets.
3. The baseline models used in numerical experiments are just k-means and stochastic block models, which are rather low bars. Other basic baselines include exponential random graphs, latent space models, random dot product graphs. There are also a wide literature of modern models in the past few years.
4. The concept of multiplex graph is closely connected to the terms "population of networks", "replicated networks", and it would be nice to discuss in this more general family of methods.

**Questions:**

1. The variational EM algorithm only guarantees to converge to a local minimizer, and the optimization objective is seen to be highly non-convex. How different are the minimizers that the algorithm converge to at different runs? And how would you interpret these different minimizers?
2. In the case of graph related modeling, it often appears that a simple spectral initialization can already give a pretty decent result. How much improvement is actually achieved by using variational EM after the spectral initialization? And without using spectral initialization, is the variational EM still capable of numerically finding a good local minimizer given sufficiently many iterations? If yes, how much iterations is saved by using spectral initialization?

---

> ### Author Response · Authors · 2023-11-22
>
> We value your insightful review feedback. All your comments have been carefully considered, and we have implemented the necessary revisions to the paper based on your suggestions. Additionally, we have conducted further experiments by variating the number of layers and number of vertices of the multiplex graph. To the best of our knowledge, no existing real dataset has been constructed with labels for both groups and blocks. Additionally, in this particular paper, we have chosen not to address the degree-corrected model due to heavy understanding of our MSBM model, in which incorporating the degree correction would further make the model difficult to understand. However, we have indicated in the conclusion that in future works, we plan to expand the model to include the degree-corrected model and explore additional distributions.
>
> Q1: As you rightly pointed out, the function is highly non-convex, and the convergence of the algorithm relies on the initialization of assignment variables. A well-informed initialization, guided by a good understanding of the groups and blocks, can lead to faster convergence. Conversely, random initialization may require more iterations to achieve optimal clustering. In any scenario, the multilayer structure proves advantageous in the assignment process by providing more samples from the same distribution. According to the law of large numbers, this abundance of samples contributes to faster convergence compared to a single-layer block model. Any discrepancies in cluster assignment between the iteration may stem from an initialization that is distant from the local minimum of the optimal solution. Thanks to our initialization model that helps to get faster and high quality convergence. It could also arise from the approximation of the posterior distribution using the variational method. We encourage users of this model to thoroughly verify its underlying hypotheses and assess whether the data aligns with these assumptions. Successful validation of these hypotheses would enhance the model's utility.
>
> Q2: In this paper, the spectral model serves as an initialization method due to its utility and speed. In all the experiments conducted, it provides a valuable indication of the assignment variables, although not a perfect partitions with less performance after VEM iteration. Despite being a generative model capable of clustering and generating data from the same distribution, the stochastic block model, in contrast to the spectral model, may yield to better result. The difference generaly depends on the type of data and the initialization of each model, notably the spectral model. Even when the spectral model provides an optimal assignment, our model possesses the ability to generate data, which is a capability lacking in the spectral model. We also explore scenarios where the initialization is randomly generated, and the variational model consistently converges to a satisfactory local minimum, although requiring more iterations, as observed in our experiments. While there are no theoretical guarantees regarding time savings from spectral initialization, our empirical results demonstrate that it is more than 10/20 times faster. It's important to note that the actual time savings may vary depending on the structure of the data.

---

> > ### Comment · Reviewer_tJJH · 2023-11-22
> >
> > Thank you to the authors for the reply. However, I don't feel that my questions/concerns have really been addressed, and will keep the score as is.

---

### Official Review · Reviewer_gZki · 2023-11-01

**Soundness:** 3 good
**Presentation:** 3 good
**Contribution:** 3 good
**Rating:** 6
**Confidence:** 3

**Summary:**

This paper discusses the setting where we are given a multi-layer graph, where each of these layers has been generated by a stochastic block model, and some of these layers are generated by the *same* stochastic block models. Detecting this structure amounts to performing two partitioning tasks at once: on the one hand, the layers need to be partitioned into groups (to determine which layers are generated by the same stochastic block model). On the other hand, for each group of layers, the corresponding block structure (partition of nodes) needs to be detected.
The authors estimate the partition of layers into groups by Estimation Maximization and estimate the partition of nodes into blocks by Variational Estimation Maximization.

**Strengths:**

The problem setup is really interesting and seems incredibly challenging. It is impressive that the authors managed to develop a theoretically sound estimation technique for this problem that is actually able to detect this subtle structure.

The introduction is well written and pleasant to read!

**Weaknesses:**

While the introduction is pleasant to read, I felt that it did not properly prepare me for the convoluted problem setup that unfolded in Section 3. Only after reading section 3 several times and going back and forth between section 3 and the introduction, did I finally understand that we are simultaneously clustering layers into groups and nodes into blocks. While I really like the problem setup, it really needs to be explained much better in order to make the reader understand how different it is from the 'normal' clustering problem. In particular, emphasis needs to be put on the difference between groups (sets of layers) and blocks (sets of nodes).

There are some notational errors in Section 3: $\beta_1$ instead of $\beta^1$ below (2), $ln$ instead of $\ln$, $[1,L]$ (continuous interval) instead of $\{1,\dots,L\}.

Section 4 is difficult to understand and the presentation (and notation) could be improved significantly.

The experiments measure the performance by the NMI measure, which is known to be biased towards fine-grained clusterings [1]. I would recommend to either use the Adjusted Mutual Information or the Correlation Coefficient [2].

The fact that the method achieves 100% accuracy on the layer-clustering in the experiments is perhaps an indication that one should choose more challenging experiments. It is interesting to see how an imperfect layer-to-group performance impacts the vertex-to-block performance.

[1] Vinh, N. X., Epps, J., & Bailey, J. (2009, June). Information theoretic measures for clusterings comparison: is a correction for chance necessary?. In Proceedings of the 26th annual international conference on machine learning (pp. 1073-1080).
[2] Gösgens, M. M., Tikhonov, A., & Prokhorenkova, L. (2021, July). Systematic analysis of cluster similarity indices: How to validate validation measures. In International Conference on Machine Learning (pp. 3799-3808). PMLR.

**Questions:**

Does the method assume some posterior distribution for the block/group sizes? In general, I'm interested to know how it relates to Bayesian Blockmodeling [1]

In the while statement of algorithm 1, shouldn't the OR be replaced by an AND? Also, maybe use $\wedge,\vee$ instead of $\|$

[1] Peixoto, T. P. (2019). Bayesian stochastic blockmodeling. Advances in network clustering and blockmodeling, 289-332.

---

> ### Author Response · Authors · 2023-11-22
> **Reply to gZki's comments**
>
> We greatly appreciate your insightful feedback on the review. All your comments have been carefully considered, and we have implemented the necessary revisions to the paper based on your suggestions. Specifically, we have rectified minor errors, adjusted the introduction to seamlessly transition into Section 3, and rephrased the fourth section to enhance reader-friendliness. Additionally, we have incorporated additional metric (Adjacent Mutual Information) to assess clustering performance. It is worth noting that, in our scenario, the performance metrics are presented for the exact number of clusters, indicating the absence of issues related to finer-grained division. Moreover, the achievement of 100\% performance in layer-to-group mapping is attributed to the model's ability to distinctly identify layers with different blocks distributions. Although we have conducted further experiments by changing the number of layers and vertices, we acknowledge that misgrouping layers could potentially impact clustering results.
>
> Q1: In our model, we do not presume any prior knowledge regarding the size of groups. Nevertheless, we do operate under the assumption that the number of clusters is known. Upon a thorough comprehension of the Bayesian technique from the proposed paper, the author posits a prior distribution concerning the number of blocks and the size of blocks. Therefore, our MSBM has not a Bayesian Blockmodeling. However, it appears that it can be generalized to be formulated with Bayesian Blockmodeling. We think that it is a relevant question to explore in future work.
>
> Q2: Certainly, we confirm the rightness of your remark. We appreciate to bring this to our attention. We correct it and we conduct a detailed revision to rectify any errors that may have occurred in the paper.

---

> > ### Comment · Reviewer_gZki · 2023-11-22
> >
> > I thank the authors for their response. I am satisfied with the answers provided and have increased my rating accordingly.
> >
> > The paper could still be improved by performing a wider range of experiments. In particular, it would be valuable if the authors conduct experiments to investigate the limits of their method: for what MSBM parameter-choices does the performance start to degrade?
> >
> > A small comment regarding NMI and granularity: fixing the number of clusters certainly reduces the impact of NMI's bias, but it does not completely solve it. NMI will still be biased towards clusterings with higher entropy (i.e., equally-sized clusters).

---

### Meta-Review · Area_Chair_648v · 2023-12-05

**Metareview:**

The paper studies the problem of clustering multi-layered graphs. The first contribution of the paper is to introduce a new model for multi-layer graphs where all the layers are generated by the SBM and some are generated by the same SBM. The paper then presents a technique to recover community in this setting.

The paper studies an important problem and introduce some interesting ideas although it suffers from two fundamental limitations:

- the model introduced in the paper is a bit complicated and not well-motivated

- the experiment are a bit artificial and not too convincing

As a result, it is not clear if the paper would have practical impact.

Overall, the paper contains some nice new directions but it is below the acceptance bar of ICLR.

**Justification For Why Not Higher Score:**

Listed above

**Justification For Why Not Lower Score:**

N / A

---

### Decision · Program_Chairs · 2024-01-16

Reject